# VoxelFeat: Voxel-wise foundation model features

**Pascual Tejero Cervera**[1]
**Samuel Joutard**[1]
**Raphael Prevost**[1]                                    PREVOST@IMFUSION.COM
**Maximilian Pietsch**[1]                                  PIETSCH@IMFUSION.COM
[1] *ImFusion, Munich, Germany*

**Editors:** Accepted for publication at MIDL 2025

## Abstract

Foundation models like SAM2 offer rich semantic features but suffer from fixed resolution, transformer artifacts, and inconsistent representations across views, limiting their direct use in 3D applications such as image segmentation. We extend FeatUp, a multi-view self-supervised upsampling approach, to 3D by introducing explicit 3D position encodings and through-plane augmentations. Our normalizer-free NFNet-based architecture enables consistent, denoised, and resolution-agnostic feature inference in medical CT volumes. The resulting 3D-aware representation supports interactive segmentation via point-wise, local inference at native resolution.

**Keywords:** Medical Image Segmentation, Feature Upsampling, Interactive Segmentation

## 1. Introduction

Foundation computer vision models such as Segment Anything Model 2 (SAM2) (Ravi et al., 2024) have demonstrated remarkable semantic understanding capabilities in medical imaging, unlocking single- and few-shot interactive use. However, while semantically rich, their feature representations lack spatial resolution and consistency. Artifacts stem from block noise and checkerboard artifacts common in transformer architectures, and semantic noise is amplified across slightly different views, such as adjacent slices in volumetric medical data or in video sequences, hindering dense downstream prediction tasks (Dosovitskiy et al., 2020; Wang et al., 2020; Hatamizadeh et al., 2021; Lu et al., 2022; Qian et al., 2021).

Our work builds upon FeatUp (Fu et al., 2024), a framework for upsampling deep features via multi-view self-supervision. One key insight of FeatUp is that the predicted foundation model semantic features from augmented views of an image should correspond to the augmented and downsampled view of a canonical set of features defined in original (the higher resolution) image space. We use the "implicit" feature model approach to map from coordinates to denoised high-resolution vision features. This model, similar to NERF models (Mildenhall et al., 2021), compresses the canonical feature set for a specific image and its augmented views in a model. The implicit model approach requires image-specific model training but yields higher-quality features than learning an image-agnostic upsampler, and it is orders of magnitude smaller than dense feature tensors (Fu et al., 2024) and allows fast, flexible, and memory-efficient inference.

**Contributions** We propose a 3D-aware adaptation of FeatUp. Our key contribution is extending FeatUp's 2D framework to handle volumetric data and voxel-specific inference. To achieve this, we use explicit 3D coordinates as input, implicit cross-slice consistency

regularization and define the model as a smooth function in the voxel grid of the image, not limited to discrete resolutions, locations, or orientations. Furthermore, we replace the network used in FeatUp by a pointwise convolutional network without normalization layers based on residual NFNets (Brock et al., 2021) to allow batched or single-voxel inference. We evaluate our approach in a medical segmentation refinement task using linear probing, demonstrating significant improvements in feature resolution and consistency.

## 2. Methods

**Training method** For multi-view generation, we sample random 2D slices from the volume with small in- and through-plane rotations, in-plane shifts, and zooming to learn a consistent 3D feature representation. The objective function, adapted from FeatUp, includes four key components: the reconstruction loss $\mathcal{L}recon$ (mean-squared error between the target foundation model prediction and the VoxelFeat feature maps), the magnitude loss $\mathcal{L}mag$ (penalizing mismatches in feature vector magnitudes), a total variation loss $\mathcal{L}TV$ (promoting spatial smoothness in the arbitrarily oriented 2D upsampled features), and a blur loss $\mathcal{L}blur$ (penalizing high-frequency artifacts). Unlike FeatUp, we exclude CT input to VoxelFeat to avoid texture bias (see section 6.13 in (Fu et al., 2024)). Additional details are provided in section 4.1.

**Proposed VoxelFeat Model** We replace the FeatUp upsampler with a point-wise residual network that maps 3D coordinates directly into the SAM2 feature space. To enable precise local inference and real-time applications, we use a normalization-free architecture with residual connections. We capture SAM2.1-large features at native CT resolution in two models: one for high-resolution blocks at $256 \times 256$, $128 \times 128$, and one for $64 \times 64$ resolution feature block (32, 64, and 256 channels, respectively). The models' feature predictions are concatenated during inference.

**Segmentation refinement experiments** To assess the semantic meaning of refined features, we evaluate the feature maps via a refinement task of a low-resolution label map generated by the nearest-neighbor down-sampling of a ground-truth mask. We train a simple point-wise linear probe with the low-resolution (noisy) mask as the target and evaluate it against the ground truth mask. Training of the segmentation head is limited to voxels within a bounding box of the noisy mask.

The segmentation head uses either the concatenated CT intensity and VoxelFeat predictions (353 channels per voxel, denoted **VoxelFeat-Probe**) or the intensity and linearly upsampled raw SAM2 vision features (**SAM-Probe**). As references, we use SAM2.1-large with two prompting strategies: (**SAM-Autoprompt**) point prompts are generated from the noisy mask in each slice (positive inside mask, negative near edges), and (**SAM-Video-Prop**) using a central slice prompt with point prompts and bidirectional propagation using the video mode. As a further reference independent of SAM2 features, we smooth the low-resolution mask with a Gaussian kernel ($\sigma = 3mm$) and binarize it (**Anti-Aliasing**), effectively performing anti-aliasing and upsampling. This baseline does not suffer from ground-truth label noise and limits discrepancies to the vicinity of the original mask boundary, except where isolated voxels were lost during downsampling. Predictions are not post-processed, but the evaluation of all methods is limited to the bounding box of the ground-truth mask dilated by $\pm 5$ voxels to limit the domain to label refinement. We evaluated these

methods on 14 CT imaages from TotalSegmentator 2.0.5 (Wasserthal et al., 2023) including labels for left kidney, L1 vertebra, spleen, right autochthon, liver, and colon (see figure 2), and on 6 CT images from KiTS23 (Heller et al., 2023) with kidney and tumor labels (see figure 3). Metrics are Dice similarity coefficient, 95th percentile Hausdorff distance (HD95), and the average distance between ground-truth and predicted mask ($\delta_{\mathrm{GT}\rightarrow\mathrm{Pred}}$) and its complementary metric ($\delta_{\mathrm{Pred}\rightarrow\mathrm{GT}}$).

## 3. Results and Discussion

| CT Volume | SAM2 PCA 1–3 | SAM2 PCA 4–6 | VoxelFeat PCA 1–3 | VoxelFeat PCA 4–6 |

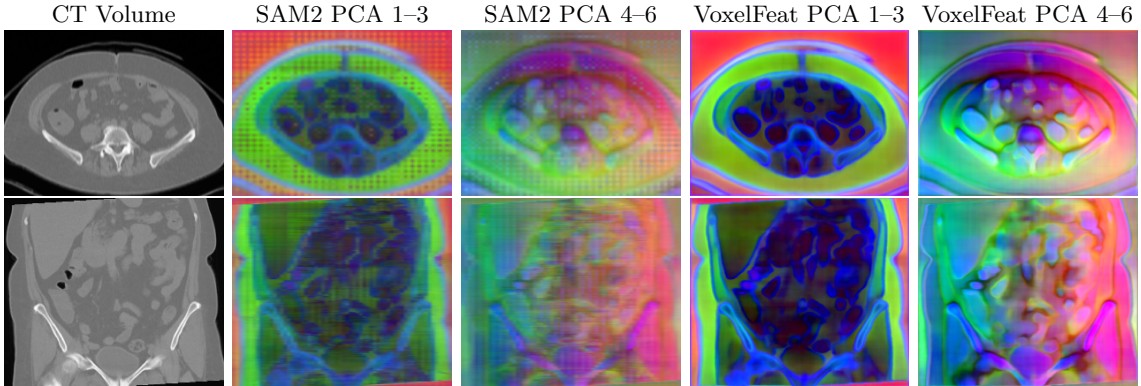

Figure 1: Exemplary TotalSegmentator data and PCA-transformed feature maps.

VoxelFeat, similar to FeatUp, produces high-resolution semantic feature maps with visibly reduced artifacts and sharp borders compared to upsampled SAM2 features. Please see figure 1 and section 4.4 for examples.

Based on surface distances and Dice scores, VoxelFeat-Probe and SAM-Probe approaches perform similar, indicating that VoxelFeat captures most semantic information but learned feature upsampling and denoising does not significantly benefit the segmentation refinement tasks in our point-wise linear probing task. VoxelFeat-Probe yields the highest Dice values for the spleen, liver, and right autochthon and all kidney and kidney tumor masks with mean Dice scores between 0.87 and 0.95. The feature probing approach underperforms compared to slice-wise prompting for vertebra and colon labels presumably due to a spatially ambiguous (adjacent vertebrae) and inconsistent feature space (background ramp, heterogeneous image appearance in the colon). Examples are shown in figures 4, 5). For label- and method-specific quantitative segmentation refinement results for both datasets and all metrics, see tables 1 and 2.

**Conclusion** In this work, we introduced VoxelFeat, a method for learning spatially consistent foundation model vision features for 3D medical data. VoxelFeat networks visually improve feature quality and allow point-wise inference, making them suitable for real-time 3D applications. However, feature noise and resolution appear not to be the limiting factors in our linear probe medical image segmentation task. These findings highlight both the promise and the current challenges of using vision feature representation for medical image segmentation.

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

## 4. Appendix

### 4.1. Objective function

The objective function is a L2-based reconstruction loss with regularization and follows closely the loss of FeatUp (Fu et al., 2024). The main reconstruction term is the (weighted) L2 norm between target foundation model feature channel $f_i(\hat{\boldsymbol{x}})$, predicted from the augmented image via the frozen foundation model, and the predicted featup feature maps $F(\hat{\boldsymbol{x}})_i$ at corresponding spatial locations $\hat{\boldsymbol{x}}$ in (augmented) low-resolution feature space. The total variation loss and blurring loss regularize high-frequency noise in the image plane, which in our case, is sampled at oblique angles of the 3D medical image grid.

We use explicit coordinate mappings to facilitate efficient resampling, which also allows our adapted reconstruction loss $\mathcal{L}_{\text{recon}}$ to be locally weighted by $w_i(\boldsymbol{x})$, the normalized expected feature channel variance $\sigma_i(\hat{\boldsymbol{x}})$ of the foundation model against features from a reference view $\hat{f_i}$ to reduce the influence of high-variance semantic noise. Furthermore, we use resampling instead of pooling operations, allowing us to define the model in the native CT space. To prevent model degradation due to non-injective mapping between (padded) coordinates (or VoxelFeat predictions) and foundation model predictions on padded images, the loss is only computed in voxels that were not padded.

$$\mathcal{L} = \underbrace{\sum_i \sum_{\hat{\boldsymbol{x}} \in \hat{X}} \frac{\|F(\hat{\boldsymbol{x}})_i - f_i(\hat{\boldsymbol{x}})\|^2}{w_i(\hat{\boldsymbol{x}}) + \epsilon}}_{\mathcal{L}_{\text{recon}}} + \lambda_{\text{mag}} \underbrace{\sum_i \left(\|F(\hat{\boldsymbol{x}})_i\|_2 - \|f_i(\hat{\boldsymbol{x}})\|_2\right)^2}_{\mathcal{L}_{\text{mag}}}$$

$$+ \lambda_{\text{TV}} \underbrace{\sum_{\boldsymbol{x} \in X} |\nabla_x F(\boldsymbol{x})| + |\nabla_y F(\boldsymbol{x})|}_{\mathcal{L}_{\text{TV}}} + \lambda_{\text{blur}} \underbrace{\sum_{\boldsymbol{x} \in X} \|\text{LowPass}\left(F(\boldsymbol{x})\right) - F(\boldsymbol{x})\|^2}_{\mathcal{L}_{\text{blur}}}. \quad (1)$$

$$\text{where} \quad w_i(\hat{\boldsymbol{x}}) = \frac{\sigma_i(\hat{\boldsymbol{x}})}{\langle \sigma_i(\hat{\boldsymbol{x}}) \rangle + \varepsilon}, \quad \sigma_i(\hat{\boldsymbol{x}}) = \sqrt{\text{E}\left[\left(f_i(\hat{\boldsymbol{x}}) - \hat{f_i}\right)^2\right]}, \quad \varepsilon = 10^{-6}.$$

### 4.2. Training method

The VoxelFeat model has 24 million parameters and was trained using the NAdam optimiser for 300 epochs with a learning rate of 0.002, with 10 epochs linear warmup followed by cosine decay. Each epoch involved 200 randomly augmented slices generating SAM2 vision features, with loss weights set at 0.001 for the magnitude loss term, 0.001 for the total variation term, and 0.1 for the Gaussian blur term. Augmentations included mirror padding (up to 15 voxels), random zoom (up to 20% of the padded image size), random cropping of the zoomed and padded image, and random rotations (both in-plane and through-plane) up to 7 degrees. To stabilize training, as commonly required by normalizer-free networks (Brock et al., 2021), we use gradient clipping, learnable residual branch skipping ("SkipInit"), and activation-based gradient scaling of the residual branch (only during the backward pass to ensure perfect point-wise predictions at inference time).

The segmentation head is single-layer point-wise convolutional architecutre with layer normalization of the inputs and a sigmoid activation. It was trained on randomly sampled

2D slices for 200 epochs using AdamW optimiser with an initial learning rate of 0.01, with a cosine annealing learning rate scheduler.

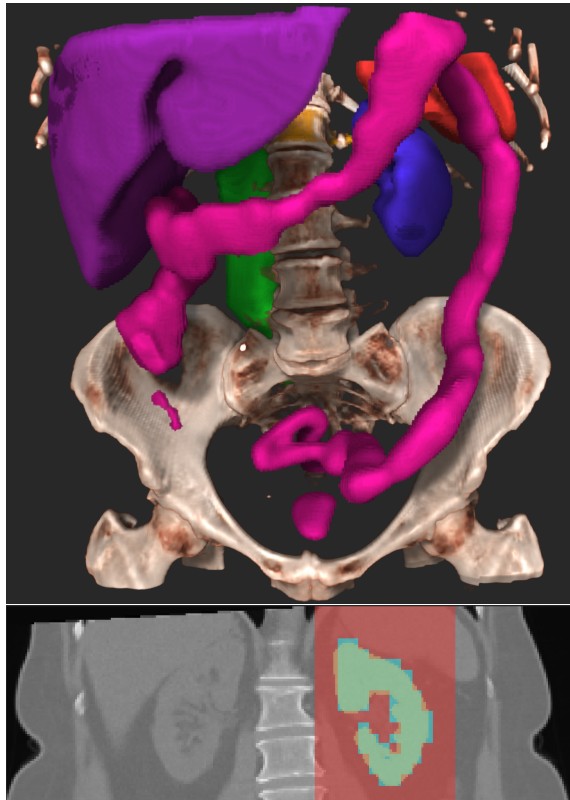

Figure 3: KiTS23 kidney (red) and tumor (blue) ground truth label maps. Both kidneys are predicted jointly.

Figure 2: Exemplary CT and target masks of TotalSegmentator data: Top shows selected labels used for evaluation; bottom shows bounding box, low-resolution, and ground truth kidney masks.

## 4.3. Label refinement results

Table 1: Segmentation metrics on TotalSegmentator data for the anti-aliasing baseline and SAM2-based methods (mean $\pm$ std over samples). The best performance using SAM features is shown in bold. Labels: 1: spleen, 3: kidney left, 5: liver, 20: colon, 31: L1 vertebra, 87: right autochthon.

| Label | Method | Dice $\uparrow$ | HD95 $\downarrow$ | $\delta_{\text{GT}\rightarrow\text{Pred}} \downarrow$ | $\delta_{\text{Pred}\rightarrow\text{GT}} \downarrow$ |
|---|---|---|---|---|---|
| 1 | Anti-Aliasing | $0.93 \pm 0.01$ | $2.90 \pm 0.26$ | $0.94 \pm 0.06$ | $0.90 \pm 0.07$ |
| 1 | SAM-Video-Prop | $0.81 \pm 0.17$ | $23.70 \pm 15.87$ | $1.76 \pm 1.26$ | $6.35 \pm 5.57$ |
| 1 | SAM-Autoprompt | $0.88 \pm 0.03$ | $7.93 \pm 3.45$ | $0.66 \pm 0.16$ | $2.36 \pm 0.83$ |
| 1 | SAM-Probe | $0.94 \pm 0.01$ | $2.71 \pm 0.54$ | $\mathbf{0.60 \pm 0.09}$ | $0.83 \pm 0.19$ |
| 1 | VoxelFeat-Probe | $\mathbf{0.95 \pm 0.01}$ | $\mathbf{2.21 \pm 0.59}$ | $0.64 \pm 0.18$ | $\mathbf{0.69 \pm 0.20}$ |
| 3 | Anti-Aliasing | $0.90 \pm 0.01$ | $2.89 \pm 0.19$ | $0.98 \pm 0.05$ | $0.92 \pm 0.04$ |
| 3 | SAM-Video-Prop | $0.86 \pm 0.07$ | $14.99 \pm 2.48$ | $2.99 \pm 0.94$ | $1.80 \pm 1.51$ |
| 3 | SAM-Autoprompt | $0.89 \pm 0.03$ | $4.88 \pm 0.71$ | $0.86 \pm 0.13$ | $1.45 \pm 0.31$ |
| 3 | SAM-Probe | $0.92 \pm 0.01$ | $\mathbf{2.53 \pm 0.63}$ | $\mathbf{0.60 \pm 0.07}$ | $0.79 \pm 0.15$ |
| 3 | VoxelFeat-Probe | $\mathbf{0.93 \pm 0.01}$ | $2.86 \pm 1.73$ | $0.74 \pm 0.24$ | $\mathbf{0.65 \pm 0.11}$ |
| 5 | Anti-Aliasing | $0.96 \pm 0.01$ | $3.00 \pm 0.00$ | $1.00 \pm 0.08$ | $0.94 \pm 0.08$ |
| 5 | SAM-Video-Prop | $0.73 \pm 0.26$ | $51.37 \pm 37.34$ | $5.99 \pm 6.41$ | $17.16 \pm 15.76$ |
| 5 | SAM-Autoprompt | $0.91 \pm 0.02$ | $12.38 \pm 4.57$ | $1.00 \pm 0.17$ | $3.33 \pm 0.87$ |
| 5 | SAM-Probe | $\mathbf{0.95 \pm 0.01}$ | $7.53 \pm 2.62$ | $\mathbf{0.99 \pm 0.23}$ | $1.96 \pm 0.57$ |
| 5 | VoxelFeat-Probe | $\mathbf{0.95 \pm 0.01}$ | $\mathbf{4.54 \pm 1.30}$ | $1.18 \pm 0.30$ | $\mathbf{1.30 \pm 0.25}$ |
| 20 | Anti-Aliasing | $0.90 \pm 0.02$ | $2.94 \pm 0.15$ | $1.00 \pm 0.03$ | $0.95 \pm 0.02$ |
| 20 | SAM-Video-Prop | $0.43 \pm 0.20$ | $108.49 \pm 50.82$ | $25.85 \pm 18.63$ | $18.87 \pm 13.68$ |
| 20 | SAM-Autoprompt | $\mathbf{0.83 \pm 0.03}$ | $\mathbf{7.31 \pm 2.86}$ | $\mathbf{0.80 \pm 0.25}$ | $2.35 \pm 0.61$ |
| 20 | SAM-Probe | $0.76 \pm 0.10$ | $21.84 \pm 12.59$ | $1.89 \pm 0.58$ | $4.98 \pm 1.92$ |
| 20 | VoxelFeat-Probe | $0.80 \pm 0.06$ | $24.08 \pm 15.05$ | $2.11 \pm 0.76$ | $\mathbf{4.73 \pm 2.65}$ |
| 31 | Anti-Aliasing | $0.82 \pm 0.01$ | $3.08 \pm 0.15$ | $1.23 \pm 0.09$ | $1.01 \pm 0.06$ |
| 31 | SAM-Video-Prop | $0.51 \pm 0.06$ | $24.59 \pm 4.97$ | $3.18 \pm 1.12$ | $7.56 \pm 1.71$ |
| 31 | SAM-Autoprompt | $\mathbf{0.79 \pm 0.03}$ | $\mathbf{4.77 \pm 0.46}$ | $\mathbf{1.03 \pm 0.20}$ | $\mathbf{1.66 \pm 0.23}$ |
| 31 | SAM-Probe | $0.76 \pm 0.04$ | $7.34 \pm 2.23$ | $1.14 \pm 0.20$ | $1.98 \pm 0.47$ |
| 31 | VoxelFeat-Probe | $0.67 \pm 0.09$ | $9.75 \pm 2.66$ | $1.86 \pm 0.56$ | $2.52 \pm 0.64$ |
| 87 | Anti-Aliasing | $0.91 \pm 0.01$ | $3.00 \pm 0.00$ | $1.09 \pm 0.04$ | $1.05 \pm 0.04$ |
| 87 | SAM-Video-Prop | $0.62 \pm 0.21$ | $42.20 \pm 14.87$ | $4.76 \pm 3.96$ | $12.26 \pm 7.85$ |
| 87 | SAM-Autoprompt | $0.84 \pm 0.02$ | $6.91 \pm 1.62$ | $\mathbf{0.70 \pm 0.08}$ | $2.38 \pm 0.33$ |
| 87 | SAM-Probe | $\mathbf{0.90 \pm 0.03}$ | $6.09 \pm 4.78$ | $0.87 \pm 0.14$ | $1.76 \pm 0.90$ |
| 87 | VoxelFeat-Probe | $\mathbf{0.90 \pm 0.03}$ | $\mathbf{4.35 \pm 1.58}$ | $1.05 \pm 0.27$ | $\mathbf{1.41 \pm 0.37}$ |

Table 2: Segmentation metrics on KiTS23 data for the baseline and SAM2-based methods (mean $\pm$ std over samples). The best performance using SAM features is shown in bold. Labels: 1: kidney, 2: tumor.

| Label | Method | Dice $\uparrow$ | HD95 $\downarrow$ | $\delta_{\mathrm{GT}\to\mathrm{Pred}} \downarrow$ | $\delta_{\mathrm{Pred}\to\mathrm{GT}} \downarrow$ |
|---|---|---|---|---|---|
| 1 | Anti-Aliasing | $0.91 \pm 0.01$ | $6.91 \pm 1.66$ | $1.79 \pm 0.26$ | $0.95 \pm 0.06$ |
| 1 | SAM-Video-Prop | $0.90 \pm 0.05$ | $20.33 \pm 19.15$ | $1.58 \pm 0.63$ | $2.73 \pm 2.32$ |
| 1 | SAM-Autoprompt | $0.91 \pm 0.02$ | $5.77 \pm 1.53$ | $\mathbf{1.09 \pm 0.31}$ | $1.24 \pm 0.25$ |
| 1 | SAM-probe | $\mathbf{0.92 \pm 0.01}$ | $\mathbf{4.87 \pm 1.06}$ | $1.15 \pm 0.15$ | $0.97 \pm 0.32$ |
| 1 | VoxelFeat-Probe | $\mathbf{0.92 \pm 0.01}$ | $5.01 \pm 1.26$ | $1.21 \pm 0.23$ | $\mathbf{0.96 \pm 0.30}$ |
| 2 | Anti-Aliasing | $0.83 \pm 0.04$ | $3.90 \pm 0.59$ | $1.35 \pm 0.04$ | $1.08 \pm 0.11$ |
| 2 | SAM-Video-Prop | $0.75 \pm 0.08$ | $14.74 \pm 4.63$ | $1.06 \pm 0.21$ | $4.29 \pm 1.85$ |
| 2 | SAM-Autoprompt | $0.84 \pm 0.03$ | $4.60 \pm 1.07$ | $0.84 \pm 0.19$ | $1.41 \pm 0.19$ |
| 2 | SAM-Probe | $0.86 \pm 0.04$ | $3.50 \pm 0.84$ | $0.86 \pm 0.12$ | $0.91 \pm 0.17$ |
| 2 | VoxelFeat-Probe | $\mathbf{0.87 \pm 0.03}$ | $\mathbf{3.35 \pm 0.56}$ | $\mathbf{0.82 \pm 0.08}$ | $\mathbf{0.81 \pm 0.06}$ |

## 4.4. Qualitative results

Exemplary PCA projections of SAM2 feature maps for TotalSegmentator and KiTS23 are shown in figures 4 and 5. The first 6 components cumulatively capture 42% and 44% of the variance of the raw SAM2 predictions, respectively.

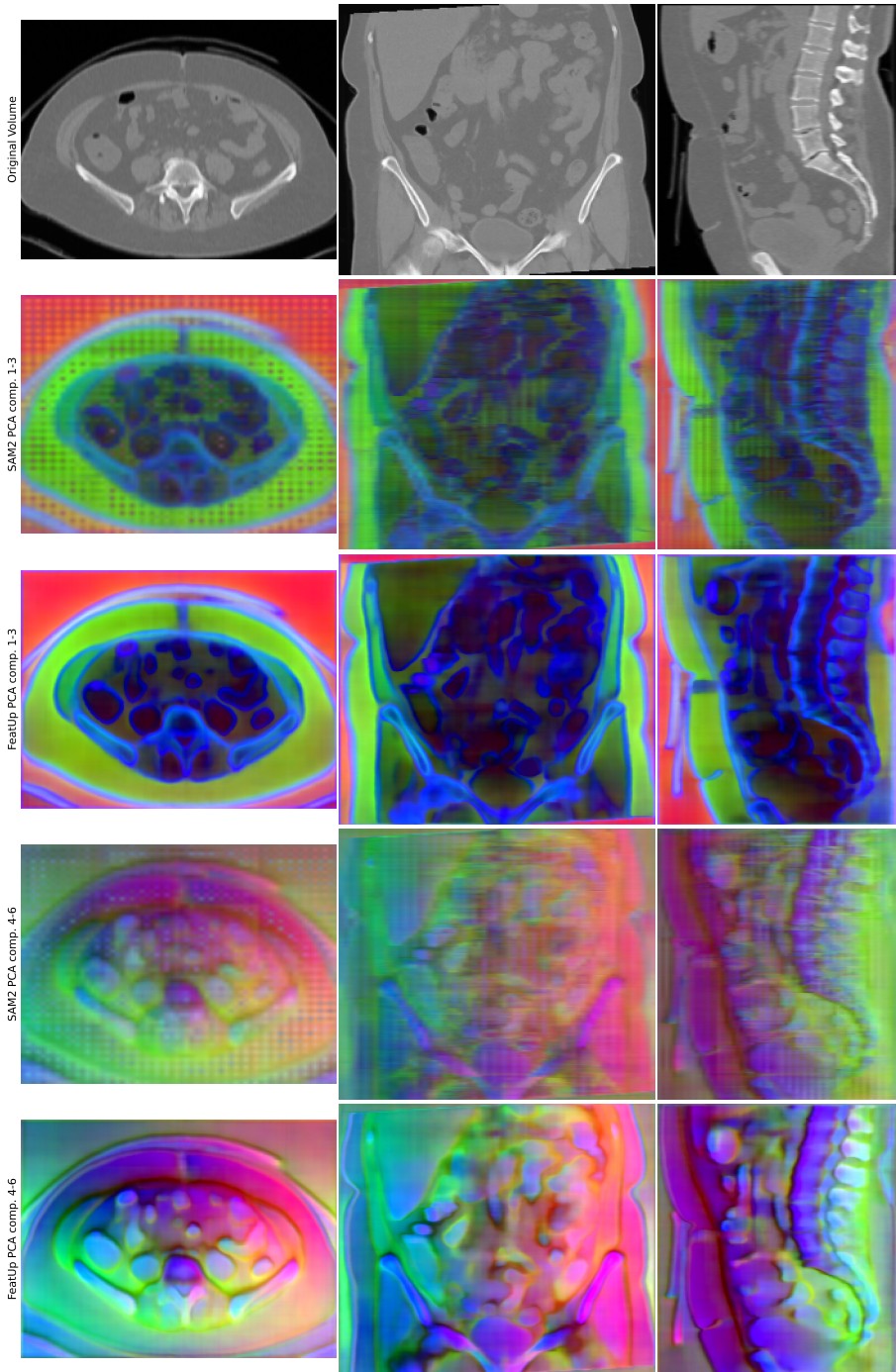

Figure 4: CT volume and first 6 PCA components of the low-res SAM2 vision features for exemplary TotalSegmentator data and corresponding VoxelFeat features projected onto the same PCA basis.

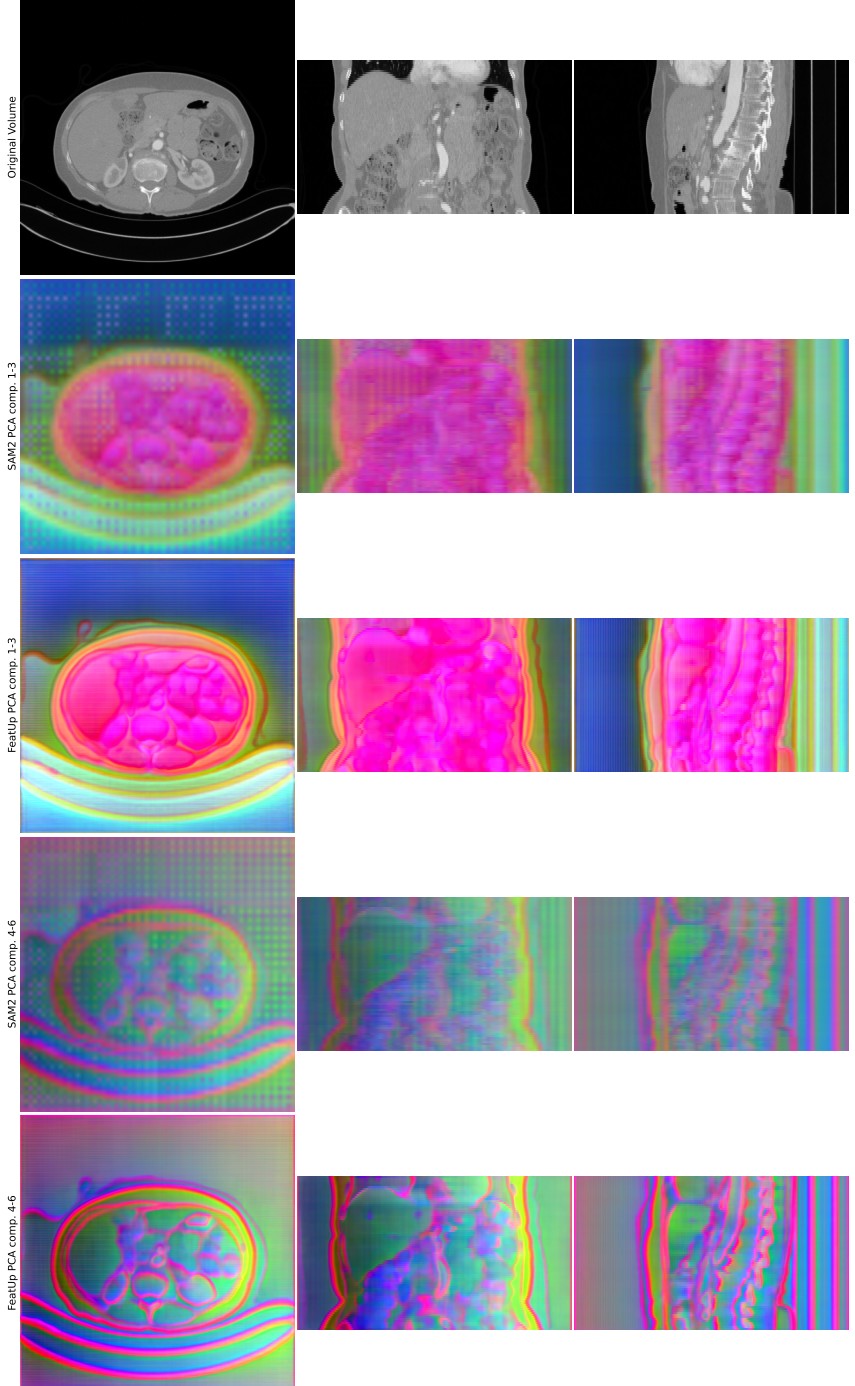

Figure 5: CT volume and first 6 PCA components of the low-res SAM2 vision features for exemplary KiTS23 data and corresponding VoxelFeat features projected onto the same PCA basis.

