# OpenReview forum: "VoxelFeat: Voxel-wise foundation model features"
_MIDL.io/2025/Short_Papers — MIDL 2025 - Short Papers_

### Official Review · Reviewer_2JSs · 2025-04-19

**Rating:** 5
**Confidence:** 5

**Summary:**

- VoxelFeat presents a method for image-specific upsampling of latent features from natural image foundation models to obtain nonlinear features on the original voxel grid.
- It is an application of the FeatUp method (Fu et al ICLR 2024) originally designed for 2D natural images to 3D features obtained by running SAM2 (a video foundation model) on slices from a volume.
- The submission also designs a strategy for efficient real-time inference.
- For its experiments, it shows applications on interactive segmentation where it does not (yet) beat SAM2, but shows promising performance.

**Strengths:**

- I quite enjoyed reading this paper and its proposed motivation is well-founded: natural vision models like SAM, DINOv2, etc. extract great features for natural images but are inapplicable to large 3D biomedical volumes. In essence, VoxelFeat comes up with a way of extending 2D methods (FeatUp and SAM2) to 3D volumes in an efficient manner at test-time and produces denoised features that are potentially usable for downstream tasks.
- I appreciate the effort to make the method fast at inference as interactive segmentation would benefit from real time applicability.
- Most obvious questions were addressed by the appendix, which is very helpful.

**Weaknesses:**

This paper well exceeds the bar for MIDL short papers, so please consider these comments more as suggestions for experiments in the eventual full paper:
- **Image-specific training requirement**: This paper requires coordinate networks to train on each image individually to get upsampled and denoised features. As the authors are already likely aware, FeatUp also has a generic methodology for upsampling features in an amortized manner without image-specific training and that should be explored in future work to make this work more broadly usable.
- **Segmentation performance**: As noted by the paper, while the upsampled features are visually appealing, they do not aid segmentation performance. There is likely a trade-off between having nice interpretable features and them being noisy and messy, such that they are nonlinear enough to aid segmentation of complicated structures. For example, a TV and a low-pass loss specifically remove high-frequency information that might be useful for segmentation. I would imagine that tuning the featup loss weights with segmentation performance as a metric would alleviate this issue.
- **Tasks beyond segmentation**: While the proposed method is not beneficial for segmentation, it is possible that the upsampled features would help on tasks such as registration. For example, [this MICCAI'24 paper](https://arxiv.org/abs/2402.15687) uses aggregated DINOv2 features for monomodal abdominal registration. As another example, alongside segmentation, [this ICLR'25 paper](https://arxiv.org/abs/2411.02372) uses its modality invariant 3D features for 3D multimodal registration.

---

### Decision · Program_Chairs · 2025-05-01

Accept